# Groove Pancreatitis—Tumor-like Lesion of the Pancreas

**DOI:** 10.3390/diagnostics13050866

**Published:** 2023-02-24

**Authors:** Gabriella Gábos, Carmen Nicolau, Alexandra Martin, Ofelia Moșteanu

**Affiliations:** 1Department of Gastroenterology, Lotus Image Medical Center, Actamedica SRL, 540080 Târgu Mureș, Romania; 2Department of Radiology, Lotus Image Medical Center, Actamedica SRL, 540080 Târgu Mureș, Romania; 3Department of Gastroenterology, Iuliu Hațieganu University of Medicine and Pharmacy of Cluj-Napoca, 400347 Cluj-Napoca, Romania

**Keywords:** groove pancreatitis, EUS-FNA

## Abstract

Groove pancreatitis (GP) is an uncommon appearance of pancreatitis represented by fibrous inflammation and a pseudo-tumor in the area over the head of the pancreas. The underlying etiology is unidentified but is firmly associated with alcohol abuse. We report the case of a 45-year-old male patient with chronic alcohol abuse who was admitted to our hospital with upper abdominal pain radiating to the back and weight loss. Laboratory data were within normal limits, except for the level of carbohydrate antigen (CA) 19-9. An abdominal ultrasound and computed tomography (CT) scan revealed swelling of the pancreatic head and duodenal wall thickening with luminal narrowing. We performed an endoscopic ultrasound (EUS) with fine needle aspiration (FNA) from the markedly thickened duodenal wall and the groove area, which revealed only inflammatory changes. The patient improved and was discharged. The principal objective in managing GP is to exclude a diagnosis of malignancy, whilst a conservative approach might be more acceptable for patients instead of extensive surgery.

A 45-year-old male smoker with a past history of severe chronic alcoholism presented to our gastroenterology department with a 2-month history of intermittent episodes of upper abdominal pain radiating to the back, nausea, postprandial vomiting, and poor appetite that persisted for 3 months, followed by a 10 kg weight loss. The patient reported no history of hypertension, previous abdominal surgery or diabetes mellitus. His family and drug history were unremarkable. Physical exams were unremarkable except for bilateral upper quadrant abdominal tenderness and hypoactive bowel sounds, while no abdominal mass was identified. Laboratory results showed that hemogram, amylase, lipase, albumin, renal and liver function tests were within normal limits. A tumor marker test found a slightly increased level of carbohydrate antigen 19-9 (CA 19-9) at 40 U/mL (normal range ≤ 30 U/mL). Carcioembryonic antigen (CEA) and alpha-fetoprotein (AFP) levels were both normal. US demonstrated mild hepatic steatosis with no cholelithiasis or acute cholecystitis but revealed general thickening of the second part of the duodenum and voluminous pancreas head. Esophagogastroduodenoscopy showed a narrowed second part of the duodenum due to an irregular, edematous, “reddish” polypoidal-appearance mass rising at the D1–D2 junction with intact extending mucosa (Figure 1). Additionally, histological examination of the pseudo-polypoid biopsy specimen revealed chronic and active mucosal inflammation and edema in the mesenchyme and was negative for malignancy. A computed tomography (CT) of the abdomen, pre-contrast phase (Figure 2), arterial phase (Figure 3) and portal venous phase (Figure 4) revealed duodenal wall thickening with luminal narrowing. It was noted that the contrast intake was parenchymal in nature and relatively homogeneous without any accompanying cystic forms. The periduodenal adipose tissue and the area of the duodenal–pancreatic groove were infiltrated with minimal adjacent fluid. Minimum densification of the right anterior pararenal fascia was observed. The cephalic pancreatic area was slightly swollen but with a relatively homogeneous acinar structure. No parenchymal calcifications or cysts were visible. The body and tail of the pancreas were healthy. The Wirsung duct and biliary system were normal as well. The pancreaticoduodenal artery was permeable and interposed between the head of the pancreas and the thickened duodenal wall. Ascites were not reported, but some pericephalic pancreatic and periduodenal lymphadenopathy, likely inflammatory, was seen.

Nonetheless, there was still a concern of malignancy due to the position of the mass. Endoscopic ultrasound (EUS) was performed (Figure 5), which described mass-like growth of the pancreatic head, narrowed duodenal wall and associated stenosis, but revealed no common bile duct (CBD) stricture or dilatation of the pancreatic duct system. EUS-FNA was performed from the exceptionally thickened duodenal wall and the groove area, which showed exclusively inflammatory changes, but no malignant or dysplastic cells. The imaging appearances (US, CT, and EUS), clinical presentation, medical history of alcohol abuse, laboratory markers and cytology results of EUS FNA were highly suggestive of GP, so major unneeded surgery was avoided in this early phase of the disease. In the absence of extreme complications (biliary obstruction or crucial gastric outlet obstruction), our patient was treated by conservative medical measures (proton pump inhibitors (PPI) and pancreatic enzyme supplement, as well as avoidance of alcohol). After being released from the hospital, the patient stopped drinking, followed a low-fat diet, and experienced no further symptoms for six months.

First described by Becker and Mischke in 1973, GP is an infrequent and still under-recognized type of recurrent or chronic pancreatitis that involves the anatomic space between the head of the pancreas, the common bile duct (CBD) and the duodenum, the so-called groove area [1]. Becker defined two forms of GP: “segmental” and “pure”. The first affects the pancreatic head with development of scar tissue within the groove, while the second involves exclusively the groove itself, sparing the pancreatic head [1].

Diagnosis is frequently challenging, and many physicians are not familiar with the disorder, which possibly contributes to its low incidence [1,2]. The accurate cause of this disorder has yet to be determined. Blockage of the minor papilla is one of the discussed aspects. Brunner gland hyperplasia is similarly thought to be a source, with stasis of pancreatic enzymes in the dorsal pancreas. Heterotopic pancreatic alterations undergoing fibrosis and inflammation in the groove area have been implicated. The most essential association is described to be an extended history of alcohol consumption. Continuous alcohol intake intensifies protein volume, which causes an escalation in pancreatic fluid thickness, provoking the inflammatory response [1,2]. 

In various studies, no difference was found in age and gender dispersion among GP and common chronic pancreatitis [1,2]. GP is generally recognized in middle-aged men with a history of significant alcohol abuse [2,3]. Clinically, patients present with chronic intermittent post-prandial abdominal pain similar to chronic pancreatitis. Some of them may present recurrent nausea, postprandial vomiting, frequently severe weight loss from impaired intestinal mobility, and duodenal stenosis [2,4]. Jaundice is infrequent in GP, contrary to pancreatic carcinoma, which presents with progressive jaundice. The duration of the clinical symptoms fluctuates from a few weeks to more than one year. The course of the GP is often chronic and debilitating [2,3,4,5]. 

Laboratory data often show little elevation of serum pancreatic enzymes and periodically of serum hepatic enzymes. Bilirubin levels can be high if the CBD is obstructed, and alkaline phosphatase levels can also be elevated despite the nonappearance of ductal reduction. CEA, AFP and CA 19-9 tumour markers are barely elevated [2,6,7]. 

Imaging plays a fundamental aspect in recognizing this entity. The literature on sonography has barely reported the appearance of GP. US commonly reveals a hypoechoic mass with thickening of the duodenal wall [5,6].

CT scans generally show mural thickening of the duodenal wall or a hypodense, insufficiently enhanced mass between the head of the pancreas and a thickened wall of the duodenum. Supplementary data include distension of the head of the pancreas and irregular calcifications. CBD may be narrowed with a smooth, tapered, and constant stenosis [8,9,10]. 

The most typical finding on magnetic resonance (MR) is a sheet-like mass corresponding to the fibrous scar in the groove among the head of the pancreas and the duodenum. MR imaging generally presents a hypointense mass on T1-weighed MR images in comparison with the pancreatic parenchyma and is iso- or slightly hyperintense on T2-weighed MR images. By contrast, administration enhancement is principally postponed due to the presence of fibrous tissues. Cystic lesions of the groove or duodenal wall may be noticed, especially on T2-weighted images. Duodenal wall thickening and duodenal wall stenosis are also commonly observed [11]. Irie et al. and Ferreira et al. reported the MRI features of patients with GP with the above MRI findings, and histological analysis revealed that these imaging features correlated with fibrous scarring in each patients [12,13].

Magnetic resonance cholangiopancreatography (MRCP) helps separate GP from CBD carcinoma, as GP shows smooth CBD tapering and shouldering is uncommon [5,8].

Esophagogastroduodenoscopy is also necessary as it can identify a congested and polypoid mucosa of the duodenum, with narrowing of its lumen or buldging of duodenal bulb [5,8,9]. Biopsies of the duodenal mucosa mostly report an incomplete result or an active inflammatory reaction without any evidence of neoplastic lesions [5,9]. Valentini et al. reported the gastrointestinal endoscopy features of their patient with GP with the above-mentioned findings [14].

The probability of accumulating samples from suspicious lesions using EUS-FNA makes EUS an ideal procedure to distinguish pancreatic adenocarcinoma from GP, allowing a diagnosis by cytopathology in approximately 90% of cases [5,9,15]. EUS can reveal narrowing and thickening of the second portion of the duodenum with intramural cysts, mild thickening of the CBD, heterogeneous hypoechoic mass and enlargement of the pancreatic head, with calcifications or pseudocysts. Regular narrowing of the CBD is seen in GP, while intermittent ductal narrowing with obstructive jaundice is seen in pancreatic adenocarcinoma. EUS FNA biopsy demonstrates enormous variability depending on the area sampled, and the presence of cytological features related to reactive cellular atypia resulting from pancreatitis may simulate malignancy [5,9,15]. To our knowledge, no studies compare EUS-FNA to FNB, specifically in GP. However, currently, Wong et al. [15] analyzed the diagnostic performance of EUS-guided tissue acquisition by EUS-guided FNA vs. EUS-guided FNB for solid pancreatic mass, and they established that the diagnostic yield of the solid pancreatic mass was higher in FNB than in FNA (94.6 vs. 89.6%). 

Radiologically, inflammatory modification in the groove between the duodenum and the pancreatic head can look indistinguishable from a malignancy. Nevertheless, it is crucial to recognize the integral clinical picture and the patient’s symptoms. A significant characteristic is the absence of major vessel encasement in GP, although some displacement may be noticed. Graziani et al. [16] described that the gastroduodenal artery is luxated leftward in GP while, in carcinoma, it is situated between the lesion and the duodenum [4,16]. Pancreatic adenocarcinoma spreading to the peripancreatic tissue or the duodenum is anticipated to penetrate and occlude peripancreatic vessels [4,17]. Ishigami et al. [18] described that patchy central enhancement in the portal venous phase is most evocative of GP, occurring in 93% of patients. Patchy central enhancement reveals pancreatic tissue in the inflammatory mass. In the same report, peripheral enhancement was only noticed in GP carcinomas. Cystic lesions in the groove are more frequent in GP than in pancreatic carcinoma [19]. A younger age is also more suggestive of GP [1,5].

Pancreatic adenocarcinomas are much more likely to invade the retroperitoneum and involve the vasculature, which was not the case with our patient. A unique finding of GP is a thickening of the medial duodenal wall, as opposed to pancreatic adenocarcinoma [2,4,5]. In our case, the diagnosis was established on clinical suspicion after a biopsy with EUS suggested an inflammatory growth. The findings that cemented the diagnosis were the lesion’s position, the luminal narrowing of the duodenum, and the minimal post-contrast enhancement of this lesion. The pancreatic duct and CBD were not enlarged, suggesting a benign nature. 

When the diagnosis is obvious, GP can be treated by conservative medical measures, including endoscopic therapy as the first line of intervention. Abstinence from alcohol, pancreatic rest, and opioid analgesics are the most commonly used conservative measures. While conservative management is preferred, resection is the gold standard in the appearance of obstructing manifestations or any suspicion of malignancy [4,5]. Therefore, it becomes essential to consider this entity as a potential and close the second differential to pancreatic carcinoma. Currently, in a review article, seven patients received endoscopic therapy, which was considered a reasonable treatment method [20]. In some studies, the primary line of management was pain management, which was mandatory in relatively half of the subjects [21]. These outcomes were identical to those found in other articles, which also revealed that conservative management was successful in half of the patients [22]. In one large retrospective case series using the endoscopic approach, linked with medical treatment, total clinical success in approximately 70% of patients was obtained in five years [23]. Still, prospective, controlled studies are needed to confirm these findings.

GP should routinely be considered in the differential diagnosis for patients presenting with pancreatic head enlargements with no cholestatic jaundice, mainly when a duodenal obstruction is present and neither duodenal biopsies nor pancreatic head FNA establishes adenocarcinoma. 

It is fundamental for physicians to become more acquainted with clinical, paraclinical and imaging findings that are evocative of GP because it can imitate pancreatic malignancy, whose prognosis and management are entirely different. Therefore, this report aims to make this entity and hidden anatomical area more recognizable to clinicians, creating a conclusive imaging diagnosis and decreasing further diagnostic work-up such as unnecessary surgeries and delayed diagnosis. 

## Figures and Tables

**Figure 1 diagnostics-13-00866-f001:**
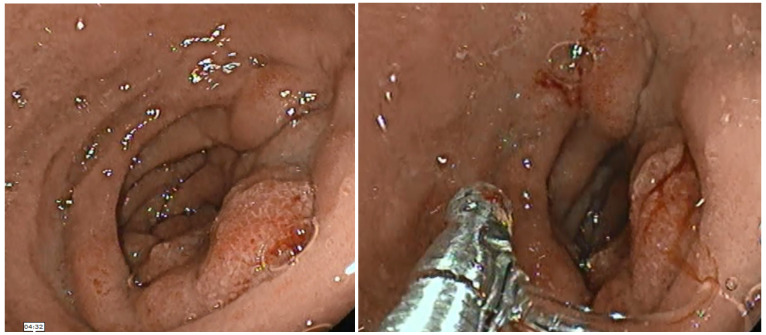
Esophagogastroduodenoscopy revealing an edematous mucosa with a pseudo-polypoid appearance narrowing the second portion of the duodenum.

**Figure 2 diagnostics-13-00866-f002:**
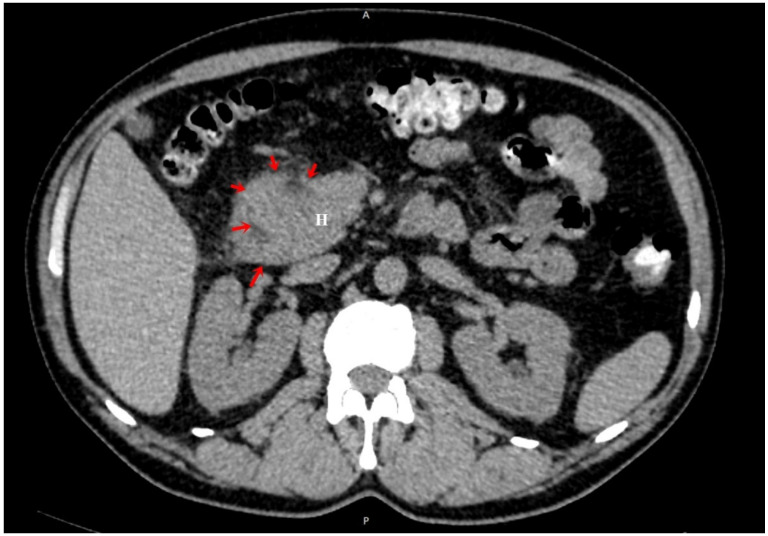
Unenhanced abdominal CT, axial plane focused on duodenum (D1–D2 segments) and pancreatic head (H) showing moderate mural thickening without delimitation from each other (red arrows). No signs of chronic pancreatitis. (A) Anterior part of the patient. (P) Posterior part of the patient.

**Figure 3 diagnostics-13-00866-f003:**
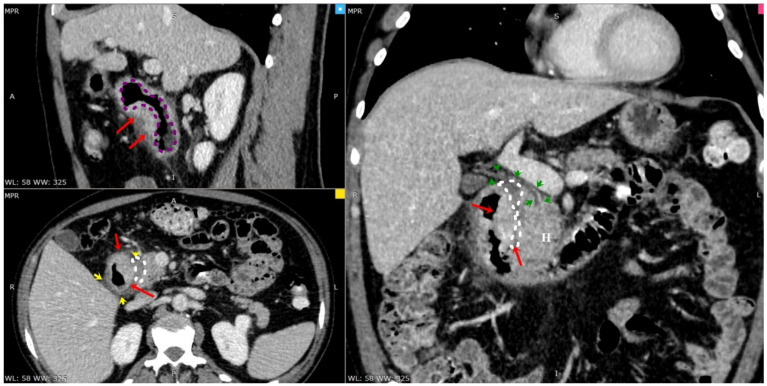
CT-MPR in the venous phase revealed duodenal lumen narrowing (purple dashed line) due to parietal thickening (red arrows) with discrete pancreatic head diffuse edema (H), a small amount of fluid surrounding duodenal-pancreatic groove (white dashed line and yellow arrows), distal Wirsung channel and distal main biliary duct (green arrowheads). No bile duct dilatation. (A) Anterior part of the patient. (R) The right of the patient. (P) Posterior part of the patient. (L) The left of the patient.

**Figure 4 diagnostics-13-00866-f004:**
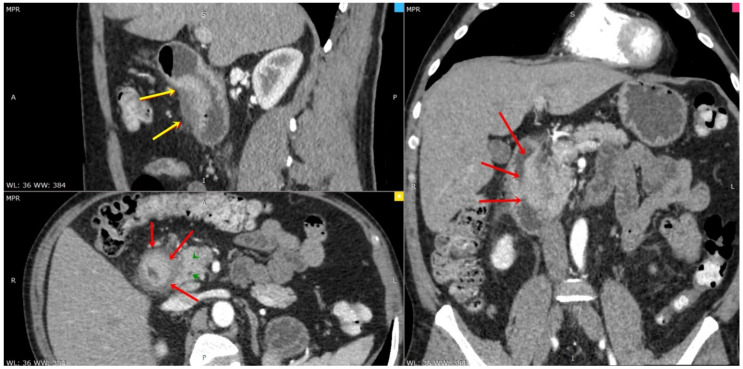
CT multiplanar reconstruction (CT-MPR) in the arterial phase revealed extended thickening of the duodenal wall (red arrows) and fluid wrapping duodenal and duodenal–pancreatic groove (yellow arrows) without signs of chronic pancreatitis or involvement of bile ducts (main biliary duct and Wirsung duct, green arrowheads). (A) Anterior part of the patient. (R) The right of the patient. (P) Posterior part of the patient. (L) The left of the patient.

**Figure 5 diagnostics-13-00866-f005:**
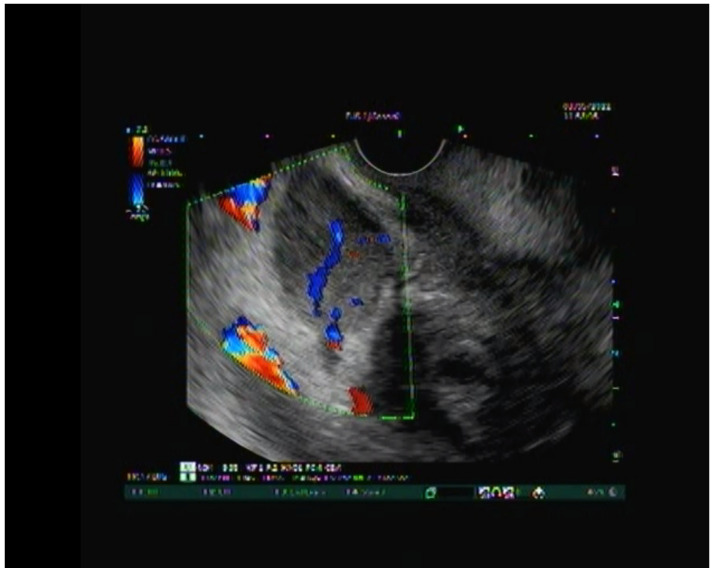
EUS—mass-like enlargement of the pancreatic head.

## Data Availability

The data presented in this study are available in the article.

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
