# Peer review of "Groove Pancreatitis—Tumor-like Lesion of the Pancreas"

_diagnostics, 2023, doi:10.3390/diagnostics13050866_

Round 1
Reviewer 1 Report
The authors present the manuscript titled "Groove pancreatitis-tumor-like lesion of the pancreas" for review. While groove pancreatitis is not a commonly encountered entity it is gaining greater recognition as a mimic of pancreatic ductal adenocarcinoma. The authors do present a report that is relatable to many in everyday practice and would be a useful reference. Below are some edits that the manuscript may benefit from:
1) Line 27: please change "tightening" to narrowing
2) Line 27: A space should be placed after the last period as a new paragraph is starting.
3) Line 86: please change to: allowing a diagnosis by cytopathology in approximately 90% of cases.
4) Line 100: please change "implicate" to involve
5) Line 102: "which is exceptional with pancreatic adenocarcinoma" this sentence structure is awkward and confusing, needs to be modified.
6) Line 104: please change "tightening" to "narrowing"
Author Response
Good afternoon!
Please see below the point-by-point response to the reviewer's comment. Also I attached the modified manuscript!
Response to Reviewer 1 Comments
Point 1: Line 27: please change "tightening" to narrowing
Response 1: “tightening” was changed to narrowing
Point 2: Line 27: A space should be placed after the last period as a new paragraph is starting.
Response 2: A new paragraph was started after the last period
Point 3: Line 86: please change to: allowing a diagnosis by cytopathology in approximately 90% of cases.
Response 3: “a cytohistological diagnosis” was changed to “a diagnosis by cytopathology”
Point 4: Line 100: please change "implicate" to involve
Response 4: “implicate” was changed to involve
Point 5: Line 102: "which is exceptional with pancreatic adenocarcinoma" this sentence structure is awkward and confusing, needs to be modified.
Response 5: "which is exceptional with” was changed to “ as opposed to”
Point 6: Line 104: please change "tightening" to "narrowing"
Response 6: “tightening” was changed to narrowing
Kind regards,
Gabriella Gabos

Reviewer 2 Report
The authors reported a case with groove pancreatitis and discussed its relation with pancreases malignancy. I have some comments:
1. The authors should report other case reports in this study as a literature review.
2. A case report has been recently published on this topic; please cite it (https://doi.org/10.22516/25007440.505).
3. Please state about the clinical presentation and diagnostic points of groove pancreatitis in more detail.
4. What about ethical issues?
Author Response
Good afternoon!
Please see below the answers to your comments.
Thank you,
Kind regard
Point 1: The authors should report other case reports in this study as a literature review.
Response 1: Additional case studies were included in the text (see comments)
Point 2: A case report has been recently published on this topic; please cite it (https://doi.org/10.22516/25007440.505).
Response 2: The requested case report was cited
Point 3: Please state about the clinical presentation and diagnostic points of groove pancreatitis in more detail.
Response 3: The clinical presentation and diagnostic points were discussed in more details (see comments)
Point 4: What about ethical issues?
Response 4: We obtained informed consent from the patient and patient’s personal information and materials do not disclose patient’s privacy. See statement’s after the presentation.

Round 2
Reviewer 2 Report
Thank you for your responsive revisions.